# Steady-State Mixing State of Black Carbon Aerosols from a Particle-Resolved Model

Zhouyang Zhang[1,2], Jiandong Wang[1,2], Jiaping Wang[3,4], Nicole Riemer[5], Chao Liu[1,2], Yuzhi Jin[1,2], Zeyuan Tian[1,2], Jing Cai[1,2], Yueyue Cheng[1,2], Ganzhen Chen[1,2], Bin Wang[1,2], Shuxiao Wang[6], and Aijun Ding[3,4]

[1]Collaborative Innovation Center on Forecast and Evaluation of Meteorological Disasters, Nanjing University of Information Science and Technology, Nanjing, 210044, China
[2]China Meteorological Administration Aerosol-Cloud-Precipitation Key Laboratory, School of Atmospheric Physics, Nanjing University of Information Science and Technology, Nanjing, 210044, China
[3]Joint International Research Laboratory of Atmospheric and Earth System Sciences, School of Atmospheric Sciences, Nanjing University, Nanjing, 210023, China
[4]National Observation and Research Station for Atmospheric Processes and Environmental Change in Yangtze River Delta, Nanjing, 210023, China
[5]Department of Climate, Meteorology, and Atmospheric Sciences, University of Illinois Urbana-Champaign, 1301 W Green St., Urbana, IL 61801, USA
[6]State Key Joint Laboratory of Environment Simulation and Pollution Control, School of Environment, Tsinghua University, Beijing, 100084, China

*Correspondence to*: Jiandong Wang (jiandong.wang@nuist.edu.cn)

**Abstract.** Black carbon (BC) exerts a notable warming effect due to its strong light absorption, largely influenced by its "mixing state". However, due to computational constraints, the mixing state is challenging to accurately represent in large-scale models. In this study, we employ a particle-resolved model to simulate the evolution of BC mixing state based on field observation. Our result shows that aerosol compositions, coating thickness (CT) distribution, and optical properties of BC aerosols all exhibit a tendency toward steady-state with a characteristic timescale of less than one day, considerably shorter than the BC atmospheric lifetime. The rapid attainment of a steady state suggests that it is reasonable to disregard this pre-steady state period and instead concentrate on the average properties of BC across extensive spatial and temporal scales. The distribution of CT follows an exponential linear distribution and can be characterized by a single slope parameter $k$. This distribution is independent of the BC-core's distribution  In the model simulation, the mean CT, equivalent to the $1/k$, is 62 nm, which is consistent with the statistical results indicating a mean CT of 63 nm. Utilizing the slope parameter $k$, which effectively characterizes the CT distribution under the steady-state simplifying assumption, the BC absorption enhancement closely corresponds to the results obtained via the particle-resolved method. This study simplifies the BC mixing state description and yields a precise evaluation of the BC optical properties, which has the potential utility for modeling efforts in the refinement of the assessment of BC's radiative effects.

## 1 Introduction

Black carbon (BC) exerts notable warming effects due to its strong light absorption (Andreae, 2001; Bond et al., 2013; Gustafsson and Ramanathan, 2016; Horvath and Trier, 1993; Jacobson, 2001; McConnell et al., 2007). BC absorption is largely influenced by its "mixing state" (Bond et al., 2013; Cappa et al., 2012; Fierce et al., 2020; Liu et al., 2017, 2015; Riemer et al., 2019), a term that describes to what extent the BC component is mixed with or coated by other aerosol components (Bondy et al., 2018; Winkler, 1973). After being emitted, fresh BC can mix with other non-absorbing aerosol components, enhancing its light absorption due to the "lensing effect" (Bond and Bergstrom, 2006; Jacobson, 2001; Peng et al., 2016; Redemann et al., 2001). This leads to larger mass absorption cross-sections (MAC) compared to freshly-emitted or bare BC. Previous laboratory studies have shown that the light absorption enhancement ($E_{abs}$) of BC aerosols can vary from 1.05 to 3.50 under controlled experimental conditions, depending on BC mixing states (Jacobson, 2001; Ramanathan and Carmichael, 2008; Riemer et al., 2019; Schnaiter et al., 2005; Shiraiwa et al., 2010). Therefore, accurately characterizing the BC mixing state is crucial for the precise assessment of the climate impact of BC aerosols.

The mixing state of BC aerosols in the real atmosphere is dynamically varying due to continuous changes caused by various processes such as the emission of particles, condensation of inorganic and organic substances on the BC aerosol, coagulation, and deposition (Moteki et al., 2007; Shiraiwa et al., 2007). Some studies have attempted to represent the BC mixing state using simplified aerosol representations, such as assuming a priori internal or external mixtures of BC aerosols (He et al., 2015; Lesins et al., 2002). However, these approaches disregard the continuous changes of the BC mixing state. Other studies have used a fixed rate to describe the aging of BC aerosols from fresh BC to aged BC (Qi et al., 2017b, a), but this approach also fails to fully capture the continuous changes.

The particle-resolved Monte Carlo & Model for Simulating Aerosol Interactions and Chemistry (PartMC-MOSAIC) offers a more detailed simulation of the continuous changes in the BC mixing state (Riemer et al., 2009; Zaveri et al., 2008). This model possesses the capability to track the state of individual aerosol particles during the simulation process (Ching et al., 2018; Hughes et al., 2018; Riemer et al., 2009, 2019; Shou et al., 2019; Zaveri et al., 2010; Zheng et al., 2021). However, the direct application of particle-resolved models within large-scale chemistry transport models or Earth system models is impeded due to the high computational cost required for their high-precision particle simulation. To overcome this challenge, the approximation of the exponential linear distribution of coating thickness (CT) under BC steady-state mixing state has been recently proposed. The study of Wang et al. (2023) suggests that particles emitted into the atmosphere undergo continuous growth and deposition processes, leading to the formation of a steady state, where the CT distribution of BC aerosols across different sizes of BC cores does not change over time. This approach has the promise to simplify the representation of the continuously changing BC mixing state in models. However, the theory of BC steady-state mixing state still requires numerical validation and key scientific questions remain unresolved: (1) Do the properties of BC aerosols demonstrate a tendency towards a steady state, and how can the steady-state characteristic time be quantified? (2) Under the steady-state assumption, can the optical properties of BC be evaluated efficiently and accurately? (3) How can the steady-

state theory be applied in models, and under what specific conditions is it applicable? Additionally, the study of Fierce et al. (2016) has noted that the composition of BC aerosols is non-uniform across the distributions of BC cores, which seems inconsistent with the conclusion of exponential linear CT distribution in the steady-state theory. This apparent discrepancy has not yet been fully explained.

In this study, we determine whether BC aerosols' mixing state reaches a steady state while the population is exposed to continuous emission, growth, and deposition processes using the PartMC-MOSAIC model. The set of baseline case was based on field observations and model simulations (Ding et al., 2016; Riemer et al., 2009; Wang et al., 2017). We also examine the characteristic time of BC aerosols reaching a steady-state mixing state, the CT distribution, and the optical properties of BC aerosols under the steady state. The steady-state CT distribution of BC aerosols will be compared with the results obtained from field observation in Nanjing, China to verify the realism of the baseline case. Moreover, we explain the apparent discrepancy between the non-uniform composition of BC aerosols found in previous studies and the exponential linear CT distribution confirmed in this study. Our study aims to provide a simplified and reasonably precise approach for the characterization of BC mixing state and evaluation of the light absorption enhancement in global models or chemical transport models.

## 2 Materials and Methods

### 2.1 Description of observational data

Observational data of BC mixing states was collected from the Station for Observing Regional Processes of the Earth system (SORPES, 118°57'10" E, 32°07'14" N; 40 m a.s.l.) in Nanjing (Ding et al., 2016). Observational data from April 2020 was used in this study. The physical properties of BC aerosols were measured by single particle soot photometers (SP2, Droplet Measurement Technologies, USA). The working principle of the SP2 has been thoroughly described in earlier studies (Gao et al., 2007; Schwarz et al., 2006). The BC mass can be determined based on its proportional relationship with the peak intensity of the emitted light and the BC mass equivalent diameter can be calculated. The scattering cross-section of BC particles is calculated using the leading-edge-only fit method developed by Gao et al. (2007). Temperature and atmospheric pressure were continuously measured by the Nanjing automatic weather station throughout the year 2020, with a time resolution of 30 minutes.

### 2.2 The particle-resolved model PartMC-MOSAIC

In this study, we utilized the PartMC-MOSAIC model to verify the process of BC aerosols reaching a steady-state mixing state. This model simulates the time evolution of the BC mixing state by tracking the composition of individual computational particles during atmospheric processes (Ching et al., 2018; Shou et al., 2019; Zaveri et al., 2008, 2010). Particle size information can be calculated from the composition of individual particles assuming spherical particles. Subsequently, the BC mixing state can be derived from the aforementioned details. We utilized PartMC-MOSAIC

simulations to obtain a precise representation of the BC mixing state, forming the numerical basis for our steady-state analysis.

To investigate the evolution of the mixing state of BC aerosols in typical regions under continuous emissions and diverse atmospheric processes, we constructed a scenario that simulates an urban area using approximately 10,000 computational particles (the precise values change throughout the simulation) for ten days. The temperature and atmospheric pressure in the scenario were set to constant values consistent with the monthly average values measured by the Nanjing automatic weather station during the period from 1 April 2020, to 30 April 2020. In this study, the initial gas concentration and emission rate were established in accordance with the parameters set by Riemer et al. (2009). To reflect the typical composition of aerosols in China, the setup for the baseline case was subsequently adjusted based on the observations reported by Wang et al. (2017) and Zhang et al. (2024), as detailed in Table S1 of the Supplement. Since time-varying emissions do not affect the average behaviour (refer to the discussion in the Supporting Information of Wang et al. (2023)), for simplicity, the emissions of gases and particles have been set to constant values in the baseline case. We prescribe three different types of carbonaceous aerosol emissions throughout the simulation representing a mix of urban sources (Cao et al., 2006; Schwarz et al., 2008b), namely pure BC particles (24.6% of the emissions by number), pure organic carbon (OC) particles (72.7% of the emissions by number), and mixed BC/OC particles (2.7% of the emissions by number), assuming that their size distributions are log-normal. The pure BC and the mixed BC/OC particles were assigned a BC-core size distribution with 89 nm geometric mean diameter and a geometric standard deviation of 1.6 following the results obtained by the SP2 instrument in Nanjing from April 1st to April 30th, 2020 (Fig. S1). The mass proportions of OC and BC are set to 31.7% and 68.3%, to ensure that particles with the BC core of 89 nm have the OC coating layer with a thickness of 20 nm (Schwarz et al., 2008b). The distribution of pure OC particle emission was assigned a geometric mean diameter of 110 nm and a geometric standard deviation of 1.7 (Fierce et al., 2016). The above settings ensure a mass ratio of 1499:4241 between component BC and component OC (Cao et al., 2006), while also maintaining a number concentration ratio of approximately 9:1 between pure BC particles and mixed OC/BC particles (Schwarz et al., 2008b). The detailed setup of the baseline case is provided in Table S1 in the Supplement.

After emission, the combustion particles underwent further coating through gas-particle partitioning, and coagulation due to Browning motion, and particles were removed by deposition, resulting in a complex mixing state. To mimic actual atmospheric conditions, we set a fixed emission rate of BC particles and maintained atmospheric processes throughout the simulation. The deposition rate of particles was set at a fixed rate and gas-particle partitioning was determined by the concentration of particles and gaseous species. While the simulation includes the full simulation of gas phase chemistry, gas-particle partitioning, aerosol thermodynamics, and the properties of BC aerosols undergoing continuous changes (Riemer et al., 2019), our study only focused on the mixture of aerosol species and the CT distribution of BC aerosols. The simulated aerosol component fractions are consistent with that in the ambient environment of China, provided in the Supplement (Table S2). To illustrate the generality of the steady state, we have created twelve additional cases by varying parameters such as temperature, emissions from natural biomass burning BC and industrial BC (Bond et al., 2013; Lee, 1983; Moteki,

2023), the growth rate within the simulated area (by increasing or decreasing particle and gas emissions by a factor of five), and diurnal gas emissions condition. Detailed descriptions of these cases are provided in Sect. 1.2 of the Supplement.

## 135 2.3 Mixing state metrics

By storing the composition of individual computational particles (i.e., the full aerosol state), PartMC-MOSAIC output contains the full information on BC mixing state. However the output is exceedingly complex, requiring about $10^5$ (particles × species) of values to describe the aerosol state at a specific time, thereby posing challenges in comparing BC mixing states at different times (Riemer et al., 2019). To overcome this, we conducted dimensionality reduction on the mixing state data
by projecting the full aerosol state onto one or multiple values and assessing the mixing state through metrics. This study employed the metrics $\chi$ (Riemer and West, 2013) and $k$ (Wang et al., 2023) to quantify the BC mixing state and track the process of BC reaching a steady-state mixing state throughout the simulation. The metric $\chi$ represents the aerosol mixing state based on diversity metrics, whereas the metric $k$ originates from the particle size distribution. The characterization of the BC mixing state using the two metrics, $\chi$ and $k$, allows us to assess the steady-state phenomenon of the mixing state by
examining both the aerosol composition and the distribution of the BC coating layer.

The diversity metric $\chi$ utilized in this study was proposed by Riemer and West (2013) to characterize the aerosol mixing state of a particle population. This metric is determined by the mass fractions of each species in each particle, ranging from 0% to 100% to represent fully external to internal mixtures. In this paper, we focus on the mixing of BC and other components. So, species are separated into two surrogate species, BC and non-BC in the calculation of metric $\chi$ (Zhao et al., 2021). PartMC-
MOSAIC output can provide the mass of species $a$ in particle $i$ represented as $\mu_i^a$, for $i$ ranging from 1 to $N$ and $a$ taking values of 1 or 2, where 1 denotes BC and 2 denotes non-BC. From this basic description of the aerosol particles, we can construct all associated quantities, as described by Riemer and West (2013) and listed in Table 1.

**Table 1. Metrics of species masses in particles, adapted from Riemer and West, 2013**

| Quantity | Meaning |
|---|---|
| $\mu_i^a$ | Mass of species $a$ in particle $i$ |
| $\mu_i = \sum\limits_{a=1}^{2} \mu_i^a$ | The total mass of particle $i$ |
| $\mu^a = \sum\limits_{i=1}^{N} \mu_i^a$ | The total mass of species $a$ in the population |
| $\mu = \sum\limits_{i=1}^{N} \mu_i$ | Total mass of population |
| $p_i^a = \dfrac{\mu_i^a}{\mu_i}$ | Mass fraction of species $a$ in particle $i$ |

| | |
|---|---|
| $$p_i = \frac{\mu_i}{\mu}$$ | Mass fraction of particle $i$ in population |
| $$p^a = \frac{\mu^a}{\mu}$$ | Mass fraction of species $a$ in the population |

By utilizing the distribution of different aerosol components both within individual aerosol particles and across the entire population, we can now calculate the mixing entropies, species diversities, and the mixing state metrics, as presented in Table 2. The entropy $H_i$ or diversity $D_i$ of an individual particle $i$ quantifies the number of effective species within a particle. This metric spans from a minimum value, where $H_i = 0$ and $D_i = 1$, indicating a particle comprised solely of a single component (either BC or non-BC), to a maximum value, where $H_i = \ln2$ and $D_i = 2$, signifying a particle composed of equal

mass proportions of BC and non-BC components. Alpha diversity ($D_\alpha$) quantifies the average effective number of species per particle within a population, with values ranging from 1, indicating that each particle is composed of a single species (though not necessarily the same species across particles), to a maximum value of 2 when all particles exhibit identical mass fractions. Conversely, gamma diversity ($D_\gamma$) assesses the effective number of species within the entire population, with values spanning from 1, signifying a population consisting of a single species, to a maximum value when there are equal

bulk mass fractions of all species present. The two population diversity metrics, $D_\alpha$ (per-particle) and $D_\gamma$ (bulk) can be integrated to yield a single mixing state metric $\chi$, which quantifies the homogeneity or heterogeneity of the population. This metric spans from $\chi = 0$, indicating a fully externally mixed population where all particles are pure, to $\chi = 1$, signifying a fully internally mixed population where all particles exhibit identical mass fractions. Therefore, the metric $\chi$ in this paper can describe the mixing of BC and non-BC components. The variation of metric $\chi$ is instrumental in more effectively tracking

the process by which BC-core is coated with other non-BC components.

**Table 2. The computation of diversity metrics, adapted from Riemer and West, 2013**

| Quantity | Name | Meaning |
|---|---|---|
| $$H_i = \sum_{a=1}^{2} -p_i^a \cdot \ln p_i^a$$ | Mixing entropy of particle $i$ | Shannon entropy of species distribution within particle $i$ |
| $$H_\alpha = \sum_{i=1}^{N} p_i \cdot H_i$$ | Average particle mixing entropy | Shannon entropy of species distribution within particle $i$ |
| $$H_\gamma = \sum_{a=1}^{2} -p^a \cdot \ln p^a$$ | Population bulk mixing entropy | Shannon entropy of species distribution within the population |
| $$D_i = e^{H_i} = \prod_{a=1}^{2} (p_i^a)^{-p_i^a}$$ | Particle diversity of particle $i$ | effective number of species in particle $i$ |

| | | |
|---|---|---|
| $D_\alpha = e^{H\alpha} = \prod_{i=1}^{N} (D_i)^{p_i}$ | Average particle (alpha) species $i$ diversity | average effective number of species in each particle |
| $D_\gamma = e^{H_\gamma} = \prod_{a=1}^{2} (p^a)^{-p^a}$ | Bulk population (gamma) species diversity | effective number of species in the bulk |
| $\chi = \dfrac{D_\alpha - 1}{D_\gamma - 1}$ | Mixing state metric | degree to which population is externally mixed ($\chi = 0\%$) versus internally mixed ($\chi = 100\%$) |

In this study, we also used the metric $k$ adopted from our previous study (Wang et al., 2023) to quantify the CT distribution of BC aerosols during the simulation. Unlike the $\chi$, metric $k$ emphasizes the thickness of the BC coating layer and

characterizes the mixing state of BC aerosols based on the CT distribution. The unified framework governing the mixing state of BC aerosols reveals a consistent size distribution, indicating that the natural logarithm of the number concentration ($\ln(n(\text{CT}))$) and CT exhibit a linear relationship, regardless of the size of the BC core (Wang et al., 2023). In this study, the variable $D_c$ denotes the diameter of the BC core and the variable $D_p$ denotes the diameter of the particle, CT signifies the coating thickness of the BC particle, which is defined as $D_p - D_c$. Furthermore, $n(\text{CT})$ represents the normalized number

concentration of BC particles within the range of $(\text{CT} - \Delta\text{CT}/2, \text{CT} + \Delta\text{CT}/2]$. The normalization is relative to the total number concentration of BC particles, rendering $n(\text{CT})$ a dimensionless value. By performing linear regression, the relationship between the CT and the corresponding number concentration in logarithmic coordinates, $\ln(n(\text{CT}))$, was established and the slope $k$ was calculated by Eq. (1). The detailed methodology for data processing is provided in Sect. 2.1 in the Supplement.

$$k = \frac{\text{d} \ln(n(\text{CT}))}{\text{d CT}} \tag{1}$$

Although the value of $k$ can be calculated using Eq. (1), it is fundamentally determined by the deposition rate (Dep), or more generally, the removal rate, and the growth rate (GR), such that ($k = \frac{\text{Dep}}{\text{GR}}$) (Wang et al., 2023). The slope $k$ was subsequently adopted as a characterization parameter to assess the BC mixing state, focusing on the particle size distribution. Per-particle size for statistical analysis was calculated using the component density from PartMC-MOSAIC output (Sect 2.2 and Table

S4 in the Supplement) combined with the core-shell model (R. McGrory et al., 2022; Wang et al., 2019).

## 2.4 Light absorption enhancement of BC aerosols

The optical properties of BC aerosols were characterized using light absorption enhancement in this study, which is defined as the ratio of the MAC of coated BC to bare BC. The calculations of $E_{\text{abs}}$ were performed using the core-shell Mie method (Fu and Sun, 2001; Schwarz et al., 2008a; Toon and Ackerman, 1981). In this study, the wavelength was set to 550 nm and

the refractive indices (RI) of the BC and scattering components were set to $1.95 + 0.96i$ (Moteki et al., 2023) and $1.5 + 0i$

(Schwarz et al., 2015), respectively. Furthermore, the determination of $E_{abs}$ is undertaken through two methods (per-particle method and $k$-value method).

The per-particle method of computation entails the utilization of core-shell Mie theory to determine the optical properties of individual particles. Subsequently, statistical techniques were applied to derive the $E_{abs}$ of the entire BC population. $E_{abs}$ could be represented as Eq. (2).

$$E_{abs} = \frac{\sum_{i=1}^{\substack{\text{total number of} \\ \text{computational particles}}} c_{abs\text{-internal}}(i) \cdot n_{conc}(i)}{\sum_{i=1}^{\substack{\text{total number of} \\ \text{computational particles}}} c_{abs\text{-external}}(i) \cdot n_{conc}(i)} \tag{2}$$

where $c_{abs\text{-internal}}$ represents the light absorption coefficients of BC core with coating shell, and $c_{abs\text{-external}}$ represents the light absorption coefficients of BC core. Both the $c_{abs\text{-internal}}$ and the $c_{abs\text{-external}}$ can be calculated directly through the core-shell Mie method with the given $D_c$ and $D_p$ of each particle $i$ simulated by the PartMC-MOSAIC model.

The $k$-value method of calculating $E_{abs}$ uses simplified $D_c$ and $D_p$ distributions, which obviates the need for individual particle calculations and greatly reduces the computational burden. The calculation is predicated on the provided probability density functions of $D_c$ and equivalent $D_p$. The log-normal distribution of $D_c$ is set according to the initial emissions specified in the PartMC-MOSAIC model, with a mean diameter of 89 nm and a geometric standard deviation of 1.6 (Lee, 1983). The probability density function of equivalent $D_p$ is determined by incorporating the mean coating thickness (Wang et al., 2023), ascertained from the $k$ value, into the probability density function of $D_c$ ($D_p = D_c + 1/k$). More detailed information and calculations about the $k$-value method were illustrated in the study of Wang et al. (2023).

The per-particle method involves computing the $E_{abs}$ of the BC population by utilizing the MAC of each particle. The $k$-value method of calculating $E_{abs}$ uses simplified $D_c$ and $D_p$ distributions obtained from the steady CT distribution, which obviates the need for individual particle calculations and greatly reduces the computational burden. Since the per-particle method demonstrates high precision in determining $E_{abs}$, the accuracy of the $k$-value method in calculating $E_{abs}$ can be evaluated by comparing it with the per-particle method.

**2.5 Determination of characteristic time**

To quantify the rate at which the BC mixing state reaches a steady state, we employ the characteristic time $\tau$, as defined by the following equation.

$$\text{metric}(t) = \text{metric}(0) \cdot e^{-\frac{t}{\tau}} + \text{metric}(\infty) \cdot (1 - e^{-\frac{t}{\tau}}) \tag{3}$$

where the metric(0) denotes the mixing state at the initial state, while the metric($\infty$) denotes the steady mixing state. In this study, we use metric $\chi$ to characterize the mixing state. The steady-state characteristic time $\tau$ enables us to ascertain the timescale over which the mixing state reaches a steady state and facilitates a quantitative comparison of the differences across various cases.

## 3 Results

### 3.1 The realism of the baseline case

To verify the realism of the baseline case simulated by the PartMC-MOSAIC model, we compare representative results of aerosol component mass fractions and CT distributions obtained from SP2 observations and model simulations. The representative results of aerosol component mass fractions from observations come from Wang et al. (2017), including mass concentrations of organic matter (OM), black carbon (BC), nitrate ion ($NO_3$), sulfate ion ($SO_4$), and ammonium ion ($NH_4$). The comparison is given in Table S2, showing that the simulated results are within the range of the observations (Wang et al., 2017). As the simulation does not involve crustal materials such as Ca and Na, the result that the positive ion $NH_4$ is slightly beyond the range is reasonable. Since there are no sulfate-containing particles emitted and the sulfate production by in-cloud chemistry is not included, the primary source of $SO_4$ is the conversion of $SO_2$ gas to particles. As a result, the proportion of $SO_4$ mass is relatively small, while $NO_3$ levels are relatively high.

Then, we conduct a statistical analysis of the CT distribution based on the results from the baseline case simulation and SP2 observations in Nanjing. Figure 1 shows that the CT distribution derived from the PartMC-MOSAIC simulation closely matches that from SP2 observations, with both results following an exponential linear distribution (linear distribution on a logarithmic scale) and yielding the same slope value of ~0.016 $nm^{-1}$ for the fitting curve, which confirms the realism of the scenario simulated by PartMC-MOSAIC model.

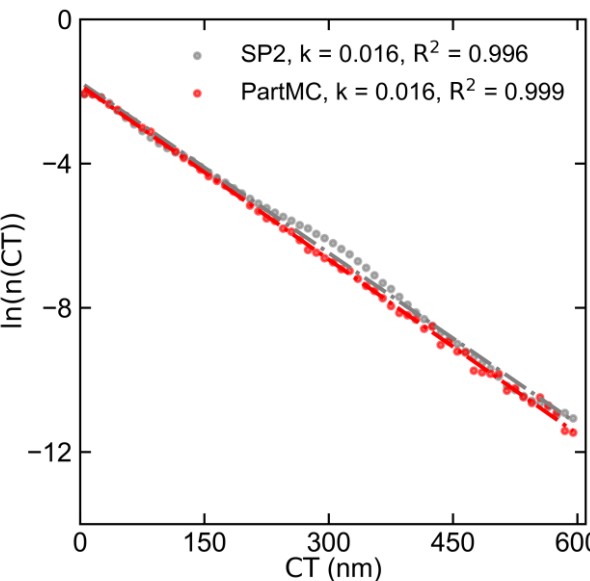

**Figure 1. The CT distribution of black carbon aerosols, determined from SP2 observation results (grey dots and line) and PartMC-MOSAIC simulation based on the observed scenario (red dots and line). Each dot indicates the total number of particles in the CT interval and the linear regression of each distribution is represented by dashed lines.**

## 3.2 The characteristic time of BC reaching a steady-state mixing state

After verifying the realism of the baseline case, we first explore whether a steady-state mixing state of BC aerosol can be achieved. Then, we use Eq. (3) to describe the progress of the mixing state metric $\chi$ reaching a steady state. The characteristic time $\tau$ for reaching a steady mixing state of the BC aerosols from the perspective of the $\chi$ criterion is investigated. We perform nonlinear fitting of the metric $\chi$ according to Eq. (3), with $\chi(0)$ given by the $\chi$ value at the start of the simulation, and $\chi(\infty)$ given by the average value of the final 48 hours of the simulation. Figure 2 illustrates the evolution of metric $\chi$ and $k$ during the simulation. We discover that the mixing state metrics $\chi$ and $k$ tend to reach a steady state, with slight daily variations (Fig. 2a and 2b) due to periodic external drivers. Thus, we adopt a moving average algorithm (Hansun, 2013) over 24 hours (Sect 2.3 in the Supplement) to eliminate such periodic influences on the mixing state. As shown in Fig. 2a, the characteristic time of the baseline case is 3.2 hours, which corresponds to the rate of the mixing state reaching a steady state. Besides, we construct more cases by changing the settings of temperature and emissions of gas and particles based on previous relevant investigations (Bond et al., 2013; Lee, 1983; Moteki, 2023). The characteristic timescale ranges from 1.9 to 9.7 hours (Table S3) for reaching a steady mixing state of the BC aerosols. Figure 2b shows the exponential linear fit slope $k$ of the CT distribution of BC aerosols. After 48 hours, the absolute value of the slope $k$ becomes steady under the periodic influence, and the value after the moving average is steady at 0.016 nm$^{-1}$. The coefficient of determination between $\ln(n(CT))$ and CT maintains a high value ($R^2 > 0.9$) during the simulation (Fig. S2), which demonstrates the stability of CT distribution over time.

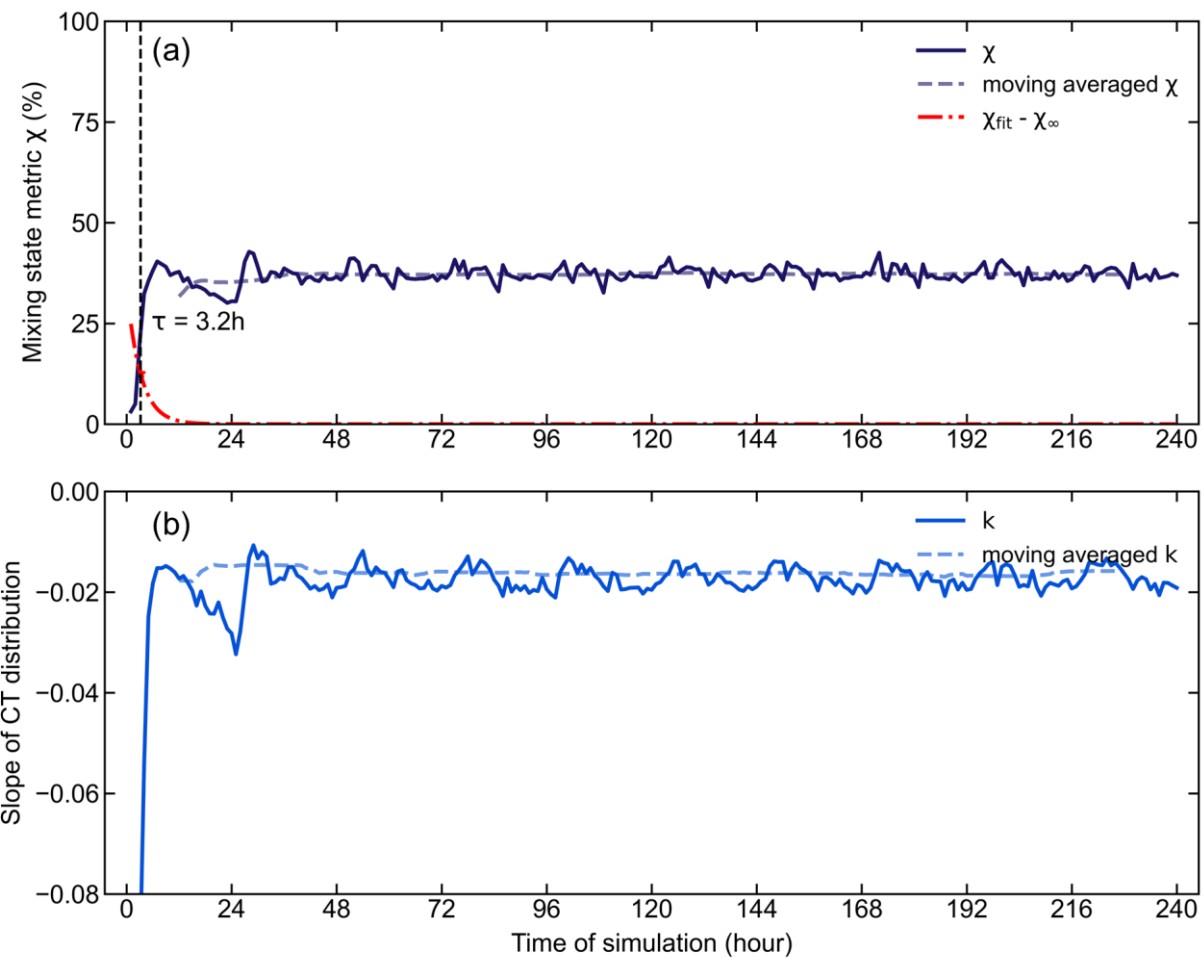

**Figure 2. The time evolution of the black carbon mixing state under the combined influence of continuous emissions and various atmospheric processes simulated by PartMC-MOSAIC during the 10-day simulation. (a) the time evolution of metric-$\chi$ (solid line), metric-$\chi$ after the simple moving average algorithm (dashed line), and $\chi_{fit} - \chi_\infty$ (red line). (b) The time evolution of metric-$k$ (solid line) and metric-$k$ after the simple moving average algorithm (dashed line).**

### 3.3 The exponential linear distribution of coating thickness and its reconciliation of non-uniform composition

Further, we analyze the CT distribution of all BC aerosols during the steady-state period (after 48 hours). Figure 1 and Fig. 3 provide the CT distribution under the steady state simulated by PartMC-MOSAIC. Figure 1 shows that the CT distribution of BC aerosols follows an exponential linear distribution. The coefficient of determination ($R^2$) between CT and $\ln(n(CT))$ is 0.999, indicating a strong linear relationship between them. The slope $k$ of the linear fitting is ~0.016 nm$^{-1}$. Equivalent to the reciprocal of $k$, the mean coating thickness is ~62 nm (Wang et al., 2023), which is only ~1 nm lower than the value calculated by the per-particle method (~63 nm). The results of CT distribution and comparison are listed in Table S3 and Fig. S3. Figure 3 shows the CT distribution results of BC aerosols, classified by the size of the BC core ($D_c$). Four $D_c$ bins (50 to

275 70 nm, 70 to 90 nm, 90 to 110 nm, and 100 to 130 nm) were selected, accounting for ~68% of the total number concentration of BC aerosols. The CT distributions with different $D_c$ have approximately the same slope of $\ln(n(CT))$ and CT, with comparable $k$ values ranging from ~0.0160 to ~0.0164 nm$^{-1}$ (Fig. 3). The above results verify that the CT distribution of BC aerosols under the steady-state condition follows the exponential linear distribution and exhibits characteristics that are not dependent on $D_c$ (Fig. 3). The exponential CT distribution of BC aerosols can effectively reflect the distribution pattern of

280 the coating layer and is more precise than simple assumptions such as the fully internal mixture of BC aerosols.

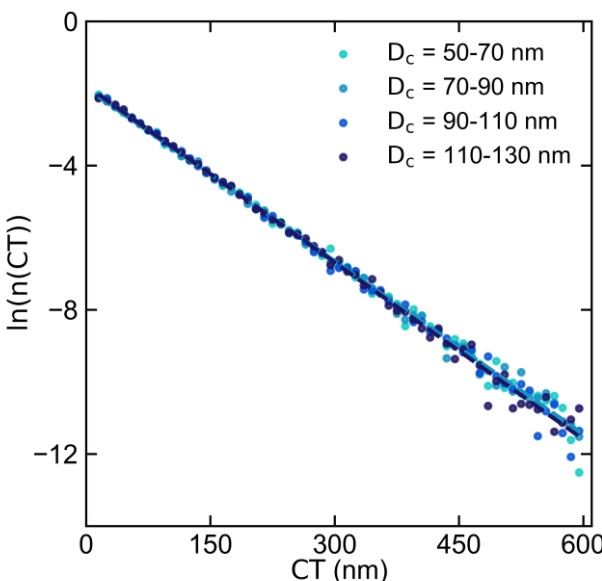

**Figure 3. The statistical CT distribution of BC aerosols during the steady-state period (after 48 hours) simulated by PartMC-MOSAIC. The CT distributions of four selected $D_c$ ranges follow the exponential linear distribution law, represented by dots with different colors. Each dot indicates the total number of particles that are in the CT interval. The linear regression of each**
285 **distribution is represented by dashed lines, with $n(CT)$ normalized by the total number concentration for each $D_c$ bin. Normalization does not influence the slope $k$ but only affects the y-intercept of the line (Sect 2.1 in the Supplement).**

Different statistical methods can reveal different facets of the BC mixing state. Fierce et al. (2016) found that across the entire diverse BC population, most of the coating material is contained in particles that have small amounts of BC mass,

while most of the BC mass resides in particles with large BC inclusions and thin coatings. The conclusion showed the non-uniform composition of BC aerosols, which seems inconsistent with the exponential linear distribution of CT independent of BC core size in our paper. Here we explain how to understand that both findings are consistent and how to utilize the exponential linear distribution of CT to parameterize the non-uniform composition of BC and non-BC components through Fig. 4, Eq. (4) and (5).

The distribution of $D_c$ follows a log-normal distribution with a mean diameter of 89 nm and a geometric standard deviation of 1.6. The distribution of CT follows an exponential linear distribution (Fig. 4a), provided by Eq. (4).

$$n(CT) = b \cdot e^{-k \cdot CT} \tag{4}$$

Where the value of $b$ is related to the total particle number concentration $N$ of BC-containing particles, and $k$ represents the slope of the exponential linear CT distribution. For statistical convenience, we normalize $n(CT)$ to make its integral over CT equal to 1, hence the value $b$ equals $k$. Because of the self-similarity of CT distributions (Fig. 3), different $D_c$ bins share the same CT distribution. Based on the distribution of CT and $D_c$, we obtain the volume fraction (VF) distribution of the BC component with respect to $D_c$, provided by Eq. (5).

$$VF(D_c) = \frac{\int_0^\infty k \cdot D_c^3 \cdot e^{-k \cdot CT} \cdot dCT}{\int_0^\infty k \cdot (D_c + CT)^3 \cdot e^{-k \cdot CT} \cdot dCT} \tag{5}$$

The derivative of the equation $VF(D_c)$ with respect to $D_c$, $\frac{dVF(D_c)}{dD_c}$, is greater than 0, which indicates that for a fixed CT distribution, VF monotonically increases with $D_c$ and BC mass of each particle. The ideal results obtained from $k$-value (exponential linear distribution of CT) corroborate the actual results from PartMC-MOSAIC simulation (Fig. 4b). Then, we use the VF of BC with respect to mass of BC contained in each particle to obtain the distribution of BC and non-BC components (Fig. 4c), which shows that components account for a smaller proportion in the range of larger BC cores and account for a larger proportion in the range of smaller BC cores, i.e., the volume (or mass) fraction of coating is not uniform across the BC cores. Based on the PartMC-MOSAIC simulation, we calculated the mass distribution of aerosol components directly and compared it with that under the uniform assumption (Fig. 4d). The results show that the distribution of aerosol components with respect to the mass of BC contained in each particle is non-uniform, and it is highly consistent with that obtained from the $k$-value in Fig. 4c, which confirms that the aerosol composition results derived from the exponential linear distribution of CT are indeed consistent with the non-uniformity result (Fig. 4d). The derivation mentioned above demonstrates that the exponential linear CT distribution of BC aerosols proves the non-uniform composition of BC aerosols from another perspective and can serve as a suitable statistical method for parameterizing non-uniform composition and characterizing the BC mixing state.

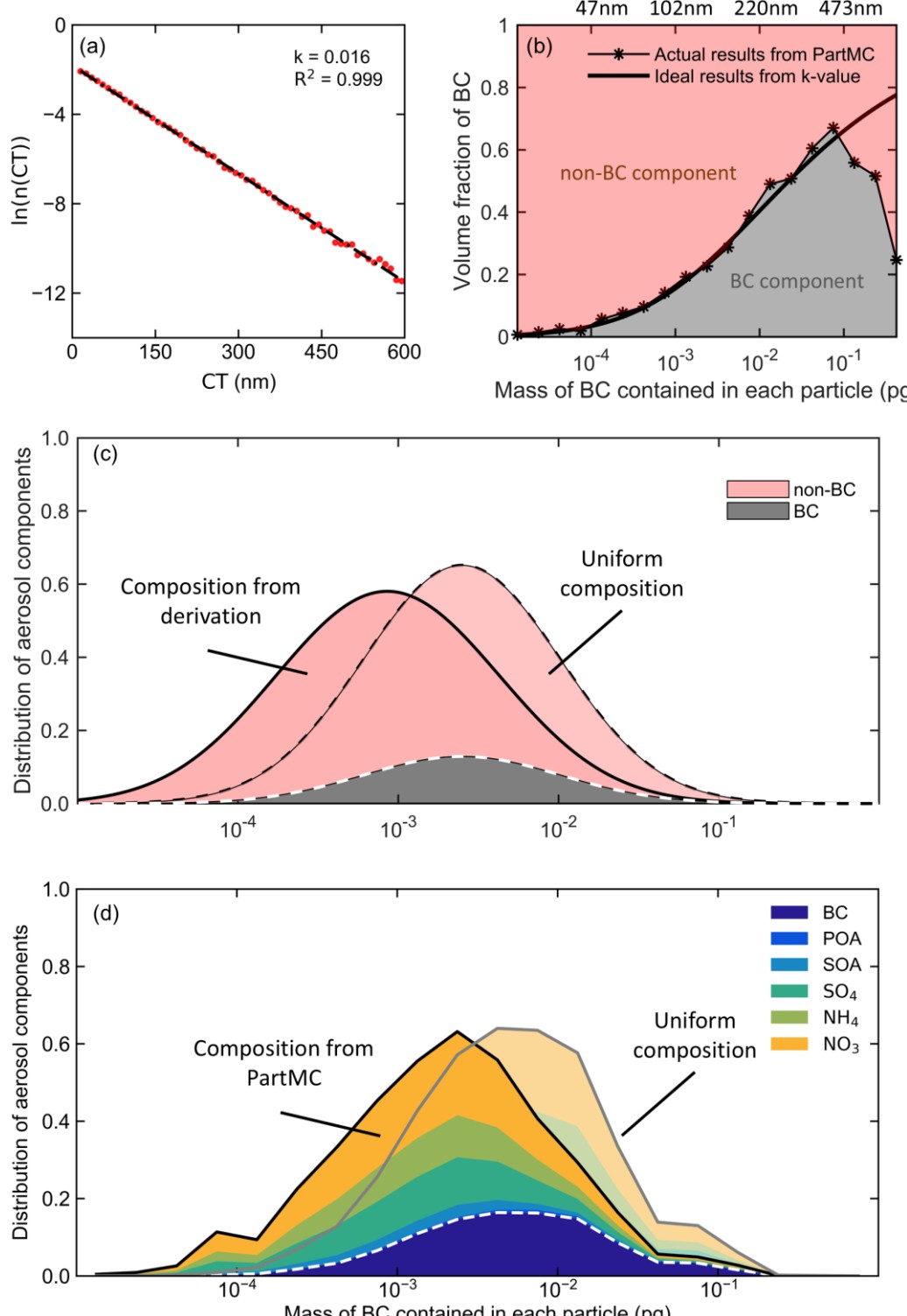

**Figure 4. The reconciliation of the exponential linear CT distribution and the non-uniform composition of BC aerosols. (a) The exponential linear CT distribution of BC aerosols with a slope $k$ of 0.016 nm$^{-1}$ under the steady state obtained from a PartMC-MOSAIC simulation. (b) The per-particle volume fraction of BC components and non-BC components with respect to per-particle BC mass. (c) The mass distribution of BC and non-BC components derived from the log-normal distribution of $D_c$ and the exponential linear distribution of CT ($k$ = 0.016 nm$^{-1}$). (d) The mass distribution of aerosol components, normalized by dividing each component's mass by the total particle mass, with respect to the BC mass in each particle, was determined using the statistical method of Fierce et al. (2017) based on the results obtained from the PartMC-MOSAIC simulation. The values on the y-axis in (c) and (d) have been normalized ($\frac{dM_i/d\log(m_{BC})}{\int_{m_{BC}=10^{-5}}^{m_{BC}=1} dM_i/d\log(m_{BC}) \cdot d\log(m_{BC})}$).**

## 3.4 The application of the $k$-value method in BC absorption

Based on the above results, we hypothesize that the exponential linear CT distribution of BC aerosols can serve as an accurate, simplified characterization of the BC mixing state under the steady-state condition. To verify its application in optical calculations, we follow the evolution of $E_{abs}$ calculated by utilizing the $k$-value method and compare it with the real result obtained from the per-particle method, as illustrated in Fig. 5. Through this comparison, we observe a minimal deviation between the $k$-value results and the true values. Specifically, the Mean Squared Error (MSE) is 0.001, the Root Mean Squared Error (RMSE) is 0.039, and the Mean Absolute Error (MAE) is 0.029. These metrics indicate that the time evolution of $E_{abs}$ from both methods shows a high degree of consistency, which verifies the viability of the method that utilizes the CT distribution to obtain the BC light absorption amplification. Additionally, Fig. 5 shows that the $E_{abs}$ reaches a steady state after ~48 hours, fluctuating regularly in the range of 1.4 to 1.5, which is consistent with observations ranging from 1.0 to 1.7 (Cappa et al., 2012, 2019; Knox et al., 2009; Lack et al., 2012; Liu et al., 2015; Ma et al., 2020; Ueda et al., 2016). Overall, the light absorption amplification of BC aerosols remains steady under the steady-state condition and the $k$-value method based on the steady-state theory can be used effectively as a simplified mixing state scheme for optical calculations of BC aerosols.

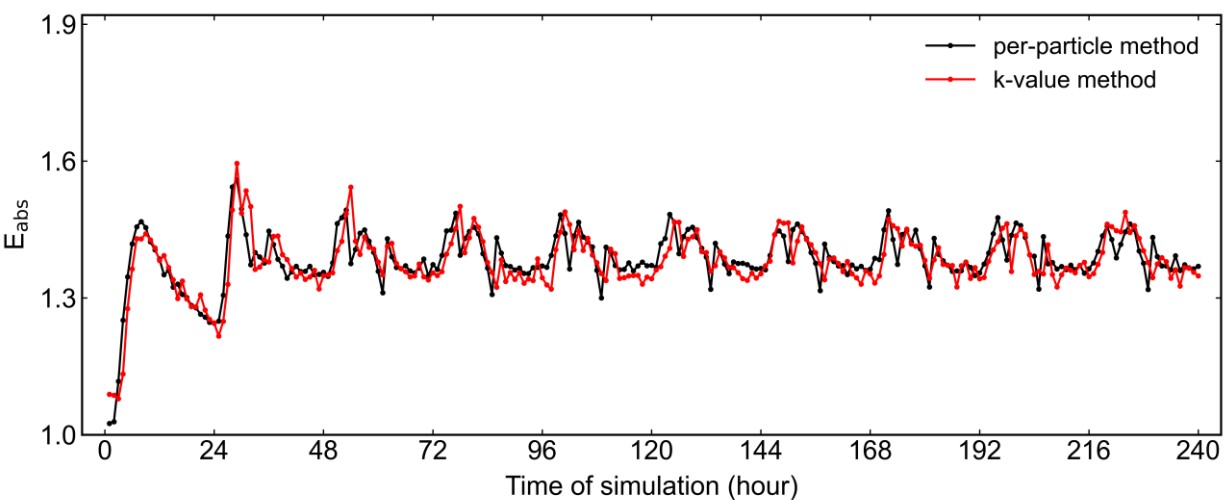

**Figure 5.** The time evolution of BC light absorption enhancement ($E_{abs}$) calculated by the per-particle method and *k*-value method during the simulation for the baseline case. The black lines and red lines represent the $E_{abs}$ calculated by the per-particle method and *k*-value method, respectively.

## 4 Discussion

We discuss the potential utility of steady-state theory for modeling efforts and its application scopes. The slope parameter *k* denotes the CT distribution of BC aerosols $n(CT) = kN \cdot e^{-k \cdot CT}$ under steady state. According to the derivation presented by Wang et al. (2023), the value of $1/k$ was determined to be equivalent to the average CT of BC aerosols (Table S3). The relationship between $E_{abs}$ and CT is approximately linear when CT is less than 200 nm ($k > 0.005$ nm$^{-1}$) (demonstrated in the Fig. S3 of Wang et al., 2023). Therefore, a monodisperse CT, 1/k, can replace the BC coating thickness distribution when calculating the BC absorption. Fig. 5 in this paper shows that the $E_{abs}$ based on the steady-state theory can serve as a characterization of the light absorption enhancement of BC aerosols under the steady-state condition. The *k*-value can be obtained from the growth rate and removal rate ($k = $ Dep/GR), thus efficiently helping evaluate the BC absorption. Regarding the application of the slope *k* in models, we propose utilizing existing state values in models, such as the total mass concentration of each component combined with the distribution of coating thickness and diameter of BC-core, to build a parameterized framework of the *k* value based on the studies by Chen et al. (2023, 2024), which is an avenue for future research. Furthermore, we plan to employ machine learning techniques to develop an emulator for the *k* value based on training data obtained from PartMC-MOSAIC. In this approach, the *k* value will serve as the "label", while emissions, initial conditions, and meteorological conditions will be treated as "features". This method is inspired by the study by Zheng et al. (2021), which used the metric $\chi$ as the "label" for their emulator. Further, the characteristic timescale for BC reaching a steady-state mixing state ranges from 1.9 to 9.7 hours, which is considerably shorter than their atmospheric lifetime,

typically around 7 days. Thus, the steady-state assumption may effectively be applied to climate models, which are generally concerned with the characteristics of BC aerosols across extensive spatial and temporal scales. It may be not applicable for high-resolution modeling studies of urban areas wherein fresh BC emissions are continuously added, which may influence the steady state. In these conditions, we propose categorizing BC-containing particles into fresh BC and aged BC (Liu et al., 2016), fresh BC follows a log-normal distribution and the coating layer of aged BC follows an exponential liner distribution. The slope parameter $k$ is capable of characterizing the distribution of the latter. Hence, the $k$-value method can be applied in models for BC absorption under the steady-state condition.

## 5 Conclusions

Our study simulated the time evolution of the BC mixing state under continuous emission, condensation, coagulation, and deposition processes using the PartMC-MOSAIC model. The baseline case is based on SP2 measurements, model simulations, and field observations, which has been verified for realism. Furthermore, based on the baseline case, we altered the pollution conditions (emission of aerosols and gas) and temperature conditions to simulate twelve additional cases. The steady-state behaviour of the BC mixing state under the joint influence of emissions and atmospheric processes was confirmed for each case. During the simulation, the characteristic time for BC aerosols to reach a steady mixing state in the baseline scenario was 3.2 hours, with a range from 1.9 to 9.7 hours under different emissions of gases and particles and different temperature conditions. In addition, the CT distributions of BC aerosols in all cases followed an exponential linear distribution. Taking the baseline case as an example, the slope of the linear fitting was 0.016 $nm^{-1}$ (with a correlation coefficient, $R^2$, of ~0.999 during steady state) and the equivalent mean coating thickness was 62 nm, close to the value (~63 nm) calculated by the per-particle method. Based on the CT distribution, we reconciled our finding with the non-uniform composition of BC particles that has been documented in previous studies. Moreover, our study found that the numerical values of $E_{abs}$ derived from the $k$-value method are highly consistent with the real values calculated by the per-particle method during the simulation. Therefore, the $E_{abs}$ based on the steady-state theory can serve as a characterization of the light absorption enhancement of BC aerosols under the steady-state condition. We successfully simplified the continuous variation of the BC mixing state and obtained a precise evaluation of the optical properties of BC aerosols based on the steady-state theory, approaching the accuracy level of the particle-resolved model. Finally, we discuss the application of the steady-state theory into global models, highlighting its application scope: specifically, the analysis of the average properties of BC across extensive spatial and temporal scales. This study has proved the importance and precision of utilizing the steady-state theory to characterize the BC mixing state and assess its optical properties, which provides a robust foundation for further research and applications.

**Code and data availability.**

The processed data, scripts, and setup of scenarios are available from the link: https://doi.org/10.5281/zenodo.13997459

(Zhang et al., 2024). PartMC version 2.5.0 was used for the simulations in this paper, available at: http://lagrange.mechse.illinois.edu/partmc/.

**Supplement.**

The supplement related to this article is available online at:

**Author contributions.**

JianW designed and directed the study. ZZ contributed to data analysis and wrote the first draft of this paper. JC, YJ, ZT, YC, and GC helped proofread the grammar of the article. JiapW, ZT and BW collected data. JianW, JiapW, NR, CL, and AD contributed the data interpretation and review of the paper.

**Competing interests.**

The authors declared that they have no conflict of interest.

**Disclaimer.**

Publisher's note: Copernicus Publications remains neutral with regard to jurisdictional claims made in the text, published maps, institutional affiliations, or any other geographical representation in this paper. While Copernicus Publications makes every effort to include appropriate place names, the final responsibility lies with the authors.

**Acknowledgements.**

We appreciate Rahul A. Zaveri for the support of the MOSAIC aerosol model. We acknowledge the High Performance Computing Center of Nanjing University of Information Science & Technology for their support of this work.

**Financial support.**

This work was supported by the National Natural Science Foundation of China (grant nos. 42422505, 42475116, and 42075098) and the National Key Research and Development Program of China (grant no. 2022YFC3701000, Task 5). N. R. acknowledges funding from DOE grant DE-SC0022130.

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
