# Peer review of "Steady-State Mixing State of Black Carbon Aerosols from a Particle-Resolved Model"

_EGUsphere, 2024_

## Author Comment (AC1)

**Response to the comments of Reviewer #1 (EGUSPHERE-2024-1924)**

*Reviewer #1: Thanks to the authors for this interesting study, which uses field observations and particle-resolved simulations to suggest that black carbon complexity can be modeled more simply than has previously been thought. Accurately quantifying black carbon's radiative impacts has proved elusive, partly because global and regional models lack the complexity needed to simulate black carbon direct effects, as pointed out by Fierce et al and others. The results of the present study look promising, and may lead to more reliable black carbon radiative impact estimates in models with parameterized aerosol properties. This is of interest largely because black carbon concentrations are extensively changing, due to regulations on industrial emissions but also increasing wildfire frequency, with current climate models having little ability to reveal how these changes affect surface temperatures. I'm hoping the authors can make some clarifications on their results and – being myself a climate modeler – I'm especially interested for more information on how these results might advance climate and chemical transport models.*

**Response:** Thanks to reviewer #1 for the great suggestions and comments. We have changed the structure of the paper and finished point-to-point responses to all comments/questions. The reviewer's comments are in italics followed by our responses and revisions (in blue). In the revised manuscript, we have positioned the reconciliation of this study's findings with those of Fierce et al. (2016) subsequent to the section on the distribution of the black carbon coating layer. Additionally, within the discussion section, we provide a detailed elaboration on the application methods and potential fields of application for the study's results. We contend that this structural arrangement enhances the coherence and logical flow of the article. Here are our point-to-point responses.

***Main comments/questions:***

***1)*** *I find it quite interesting that a steady state is reached within a day by the chosen metrics, but would like more explanation for why this particular result is useful from a modeling perspective. Would I be interpreting correctly to say that the pre-steady state period is harder to model than the steady state itself, yet is sufficiently short that an accurate representation of BC properties could reasonably ignore this period and focus on estimating steady-state chi and k values? I imagine such logic might hold reasonably for aerosols averaged over large regions (as in a GCM), but may fare less well for, say, a high-resolution modeling study of an urban area wherein a larger percent of BC emissions are fresh.*

**Response:** Thank you for your comment. Just as you mentioned here, simulating the evolution of black carbon (BC) during the pre-steady-state period may be complicated in large-scale models due to complex variations of BC's properties under the effects of various atmospheric processes. Additionally, the widely accepted atmospheric lifetime of BC is approximately 7 days, significantly longer than the 1.9 to 9.3 hours in our study. Hence, the rapid attainment of a steady state suggests that it is reasonable to disregard this pre-steady-state period and instead concentrate on analyzing the average properties of BC across extensive spatial and temporal scales. The spatial and temporal resolutions necessary for modeling differ based on the simulation objectives. Climate models, which are generally concerned with the characteristics of BC aerosols across extensive spatial and temporal scales, may employ the steady-state assumption. While the assumption may fare less well for high-resolution modeling studies of urban areas wherein a larger percent of BC emissions are fresh, in which the steady state may be influenced. In this condition, we propose categorizing BC-containing particles into two distinct types: fresh BC, which lacks a coating layer, and aged BC, which possesses a coating layer. Our findings are pertinent to characterizing the distribution of the latter. Based on the above content, we have updated the discussion section, providing a detailed introduction to methods and potential fields of application for the study's results.

We have added the related description in the revised manuscript (Section 4, Line 362-370 ):

*"Further, the characteristic timescale for BC reaching a steady-state mixing state ranges from 1.9 to 9.7 hours, which is considerably shorter than their atmospheric lifetime, typically around 7 days. Thus, the steady-state assumption may effectively be applied to climate models, which are generally concerned with the characteristics of BC aerosols across extensive spatial and temporal scales. It may be not applicable for high-resolution modeling studies of urban areas wherein fresh BC emissions are continuously added, which may influence the steady state. In these conditions, we propose categorizing BC-containing particles into fresh BC and aged BC (Liu et al., 2016), fresh BC follows a log-normal distribution and the coating layer of aged BC follows an exponential liner*

*distribution. The slope parameter k is capable of characterizing the distribution of the latter. Hence, the k-value method can be applied in models for BC absorption under the steady-state condition."*

**2)** *As I'm a modeler I'd be very interested if the authors can please further explain how their findings could be used to advance BC treatment in models. Could the authors envision simple parameterizations for the steady-state chi and k being put into a climate or chemical transport model, as functions of emission rates and conditions? Presumably this would depend on the ratio of emitted BC to organics, the total mass, and possibly meteorological conditions and other factors. I'm interested in the authors' thoughts here, and ideally they might expand their two sentences in the Conclusion (Lines 334-8).*

**Response:** Thank you for your comments. Our study corroborates that the characteristic timescale for BC approaching a steady-state mixing state is significantly shorter than its atmospheric lifetime. This finding suggests that the properties of BC under the steady state can effectively represent the averaged properties of BC across extensive spatial and temporal scales. The parameter $k$ denotes the distribution of coating thickness ( $n(\mathrm{CT}) = kN \cdot e^{-k \cdot \mathrm{CT}}$ ) under steady state. According to the derivation presented by Wang et al. (2023) and the findings in this paper (Table S3), the value of $1/k$ was determined to be equivalent to the average CT. The relationship between $E_{\mathrm{abs}}$ and CT is approximately linear when CT is less than 200 nm ($k > 0.005$ nm$^{-1}$). This linear behavior is clearly illustrated in Fig. 1, which corresponds to Figure S3 in Wang et al. (2023).

[Figure]

Figure 1. Change of mass absorption cross-section (MAC) of black carbon (BC) with coating thickness (ΔDp). Blue dots represent the calculated MAC based on core-shell Mie theory with 5 the linear fit shown as the red line.

Therefore, the BC coating thickness distribution can be replaced by a monodisperse CT, $1/k$, when calculating the BC absorption. Fig. 5 in this paper validates this simplification.

The $k$-value can be obtained from the growth rate and removal rate ($k$ = Dep/GR), thus efficiently helping calculate the light absorption capacity of BC aerosols. Regarding the application of the slope $k$ in models, we propose utilizing existing state values in models, such as the total mass concentration of each component combined with the distribution of coating thickness and diameter of BC-core, to build a parameterized framework of the $k$ value based on the studies by Chen et al. (2023, 2024), which is an avenue for future research. Furthermore, we plan to employ machine learning techniques to develop an emulator for the $k$ value based on training data obtained from PartMC-MOSAIC. In this approach, the $k$ value will serve as the "label", while emissions, initial conditions, and meteorological conditions will be treated as "features". This method is inspired by the study by Zheng et al. (2021), which used the matrix χ as the "label" for their emulator. These tasks are avenues for our future work.

We have added the related description in the revised manuscript (Section 4 Paragraphs 2, Lines 349-362):

*"We discuss the potential utility of steady-state theory for modeling efforts and its application scopes. The slope parameter k denotes the CT distribution of BC aerosols $n(CT) = kN \cdot e^{-k \cdot CT}$ under steady state. According to the derivation presented by Wang et al. (2023), the value of 1/k was determined to be equivalent to the average CT of BC aerosols (Table S3). The relationship between $E_{abs}$ and CT is approximately linear when CT is less than 200 nm ($k > 0.005$ nm$^{-1}$) (demonstrated in the Fig. S3 of Wang et al., 2023). Therefore, a monodisperse CT, 1/k, can replace the BC coating thickness distribution when calculating the BC absorption. Fig. 5 in this paper shows that the $E_{abs}$ based on the steady-state theory can serve as a characterization of the light absorption enhancement of BC aerosols under the steady-state condition. The k-value can be obtained from the growth rate and removal rate (k = Dep/GR), thus efficiently helping evaluate the BC absorption. Regarding the application of the slope k in models, we propose utilizing existing state values in models, such as the total mass concentration of each component combined with the distribution of coating thickness and diameter of BC-core, to build a parameterized framework of the k value based on the studies by Chen et al. (2023, 2024), which is an avenue for future research. Furthermore, we plan to employ machine learning techniques to develop an emulator for the k value based on training data obtained from PartMC-MOSAIC. In this approach, the k value will serve as the 'label', while emissions, initial conditions, and meteorological conditions will be treated as 'features'. This method is inspired by the study by Zheng et al. (2021), which used the matrix χ as the 'label' for their emulator."*

***3) I quite like the reconciliation of this study's results with those of Fierce et al 2016, though I feel a few lines on that study should be more accurate. For instance Line 27: "diversity in the distribution of BC coating thickness that has been documented in previous studies", and Line 62: "the study of Fierce et al. (2016) has noted that the coating thickness of BC aerosols is non-uniform across the distributions of BC cores". The Fierce et at study describes particle-resolved results as dissimilar to "uniform composition" and "uniform mixing", but not uniform coating thickness, so I find this inaccurate. Further, these descriptions make it sound very much like that study and the current one directly disagree, which is not made out to be the case. Further, this is more***

*directly indicated to be the case in Line 70 "we explain the discrepancy between [...]"* *rather than an "apparent discrepancy" as in Line 64.*

**Response:** We are pleased to note your interest in this "reconciliation" and appreciate your insightful guidance on our work. In response to your comments, we have implemented the following revisions to the manuscript: 1. Regarding the findings of Fierce et al., we have employed the term "non-uniform composition" to describe the phenomena. Consequently, due to the non-uniform composition of black carbon aerosols, the volume of coating components and black carbon component exhibits non-uniformity (i.e., the ratio of coating thickness to BC-core's diameter per particle is not a constant value). 2. We have consistently employed the term "exponential linear distribution of coating thickness" or "exponential linear CT distribution" to describe the findings of the distribution of coating thickness. 3. To articulate the relationship between the two, we utilized the phrases "seem inconsistent" and "apparent discrepancy." The conclusion showed the uniform composition of BC aerosols, which seems inconsistent with the exponential linear distribution of CT in our paper. Here we explain how to understand that both findings are consistent and how to utilize the exponential linear distribution of CT to parameterize the non-uniform composition of BC and non-BC components through Fig. 4, Eq. (3) and (4)." We have made these revisions in the manuscript, hoping that these changes will make the study more accurate and readable.

Line 69 in the revised manuscript: "*Additionally, the study of Fierce et al. (2016) has noted that the __composition of BC aerosols is non-uniform__ across the distributions of BC cores,*"

Line 69 in the revised manuscript: "*which __seems inconsistent__ with the conclusion of exponential linear CT distribution in the steady-state theory. This discrepancy has not yet been fully explained.*"

Line 78 in the revised manuscript: "*Moreover, we explain the __apparent discrepancy__ between __the non-uniform composition of BC aerosols__ found in previous studies and __the exponential linear CT distribution__ confirmed in this study.*"

Line 291 in the revised manuscript: "*The conclusion showed the __non-uniform composition of BC aerosols__, which __seems inconsistent__ with the __exponential linear distribution of CT independent of BC core size__ in our paper. Here we explain how to understand that both findings are consistent and how to utilize the exponential linear distribution of CT to parameterize the non-uniform composition of BC and non-BC components through Fig. 4, Eq. (4) and (5).*"

*4) I'm wondering if some of the terminology related to the interpretation of results could be clearer. Is "uniform" a clear enough description of the slopes closely following a k value? This could suggest something being uniformly distributed, which is not the case here (despite Fig. 5's legend mentioning a "uniform distribution"). Would "independent of BC core size" or just "size-independent" be more accurate? For the several comparisons between "uniform" and "non-uniform", these aren't innately incompatible but the wording makes it seem that this is so. Perhaps a clearer summary would just say that the coating volume fraction varies with BC core size, but coating*

*thickness is size-independent? Maybe there's a better description the authors have in mind?*

**Response:** Thank you for your correction. The coating thickness (CT) distribution in our previous manuscript version was described using the term "uniform", which led to substantial misunderstandings. To enhance clarity and facilitate a more rigorous comparative analysis, we have updated the terminology in the manuscript. The precise characterization of the CT distribution should be "exponential linear distribution." The term "linear" is particularly advantageous for correlating the slope "$k$" with the distribution. Furthermore, our initial manuscript inadvertently suggested a misleading "apparent discrepancy" between "uniform" and "non-uniform" in several comparative analyses. The explanation of the term "discrepancy" you provided, specifically that "the coating volume fraction varies with BC core size, but coating thickness is size-independent," inspired us and was particularly insightful for our research. We propose that the discrepancy and connection between the results reported by Wang et al. and Fierce et al. can be articulated as follows [Page 13, lines 316-318]:

*" the exponential linear CT distribution of BC aerosols proves the non-uniform composition of BC aerosols from another perspective and can serve as a suitable statistical method for parameterizing non-uniform composition and characterizing the BC mixing state. "*

**5)** *Related to my comment immediately above, could the authors please comment on whether the normalization used to plot the data (Fig. 3) affects interpretation of the results?*

**Response:** Thank you for your question. Normalization used to plot the data (Fig. 3) does not affect interpretation of the results. The "$n(CT)$ normalized for each $D_c$ bin" in the caption of Fig. 3 means that the number concentration of each CT bin is divided by the total number concentration for each $D_c$ bin (dots and lines of each color). Therefore, the y-axis is represented as $\ln n(CT) - \ln n(D_c)$, and the slope, denoted as $k$, is given by $\frac{d \ln n(CT)}{d\ CT} - \frac{d \ln n(D_c)}{d\ CT}$. Since the derivative with respect to CT after the subtraction is zero, normalization does not influence the slope $k$ but only affects the y-intercept of the line. To address the inconsistency in total particle number concentration across different $D_c$ bins, we implemented a normalization technique to mitigate the impact of varying particle concentrations. This approach facilitates a more accurate comparison of the coating thickness distribution across different $D_c$ bins, specifically to determine whether the slope $k$ remains consistent.

We have added the related description in the revised manuscript (Caption of Figure 3, Page 12, Lines 285-287):

*"The linear regression of each distribution is represented by dashed lines, with n(CT) normalized by the total number concentration for each Dc bin. Normalization does not influence the slope k but only affects the y-intercept of the line (Sect 2.1 in the Supplement)."*

Besides, to describe the influence of normalization on CT distribution, we add a Paragraph in sect. 2.1 in Supplement.

*"To eliminate the influence of total particle number concentration across different $D_c$ bins, we implemented a normalization technique to mitigate the impact of varying particle concentrations. This approach facilitates a more accurate comparison of the coating thickness distribution across different $D_c$ bins, specifically to determine whether the slope k remains consistent. Normalization does not influence the slope k but only affects the y-intercept of the line. The number concentration of each CT bin is divided by the total number concentration for each Dc bin in Fig. 3. Therefore, the y-axis is represented as $ln(n(CT)) - ln(n(D_c))$, and the slope parameter k is given by $\frac{d\ ln\ n(CT)}{d\ CT}$*

$- \frac{d\ ln\ n(D_c)}{d\ CT}$, *where $n(D_c)$ represent the total number concentration $\int_0^{600} n(CT)\ dCT$ in the specific $D_c$ bin. Since the derivative with respect to CT after the subtraction is zero, normalization does not influence the slope k but only affects the y-intercept of the line."*

**6)** *The results are based on field observations in Nanjing. Can the authors please comment on whether Nanjing is sufficiently representative for the key results to hold generally? Would the steady state timescale be quite different for a site with natural biomass burning black carbon, rather than industrial black carbon? I see there are some simulated cases that are variants from the Nanjing one, but I don't have a sense by how much.*

**Response:** Thank you for your inquiry. In addition to the Nanjing case, simulations were conducted for various other pollution scenarios. In the baseline scenario, field observations in Nanjing provided data solely on the distribution of $D_c$ and the prevailing meteorological conditions.

To improve the generalizability of our results, we varied the temperature, the size distribution of freshly emitted BC, and the particle-gas emission ratio, constructing nine additional cases. In response to the reviewer's comments regarding natural biomass burning black carbon, we have incorporated three additional scenarios into the revised manuscript. The geometric mean diameter of freshly emitted BC was set as 150 nm based on Fig. 2, which corresponds to Figure 4 of Bond et al.(2013), and varying particle-gas emission ratios. In these three new scenarios, the CT distribution follows an exponential linear distribution, with a steady-state timescale ranging from 2.1 to 2.8 hours, within which our results still hold. Therefore, the CT exponential linear distribution and the range of steady-state timescales (less than one day) can be considered universal.

[Figure]

Figure 2. Mass and number size distributions of BC particles observed in three fresh urban (red) and two fresh biomass burning (black) plumes as identified in the legend. The measurements are made on board an aircraft using an in situ, single-particle detection instrument (SP2). The variable coatings on the BC particles are not shown. The observed (a) mass and (b) number amounts are plotted as symbols versus volume equivalent diameter based on assuming a spherical particle shape. The mass distributions are normalized to the same peak value. The observations are fit by a lognormal function between 90 and 600 nm (solid lines). The number distribution fits are those consistent with the fit to the respective mass distribution and are scaled to represent the same BC mass. From Schwarz et al. [2008b].

The introduction of the baseline case was changed into the following one [Page 3, Lines 73-74].

*"The set of baseline case was based on field observations and model simulations (Ding et al., 2016; Riemer et al., 2009; Wang et al., 2017)."*

The three new cases are detailed in Sect. 1.2 of the Supplement, with their CT distributions shown in Figures S3 (j), (k), and (l), and steady-state characteristic timescales listed in Table S3 of Supplement.

[Figure]

*"Part of Fig S3. Results of CT distribution of BC aerosols in 12 simulation cases (excluding the Baseline case)."* [Page 7 in the Supplement]

*"Part of Table S3. The characteristic time τ, equivalent CT (1/k), coefficient of determination of linear regression, and the average true CT value for ten different cases."* [Page 8 in the Supplement]

| *Scenario* | *Characteristic time (hour)* | *Equivalent CT (nm)* | *Coefficient of determination* | *Average true CT (nm)* |
|---|---|---|---|---|
| *Baseline case* | *3.2* | *62* | *0.999* | *63* |
| *Case 10* | *2.8* | *29* | *0.995* | *29* |
| *Case 11* | *2.1* | *71* | *0.999* | *76* |
| *Case 12* | *2.2* | *157* | *0.998* | *162* |

**7)** *Could the authors please briefly explain their reasoning behind the statement that chi and k give a "comprehensive depiction of the BC mixing state" (Line 134)? Certainly these two parameters could be superior to mixing state index only, but as this is presumably the first study to combine these metrics there might not be anything to cite in support of this statement, so I request a little explanation.*

**Response:** Thank you for your inquiry. Our earlier statement, "chi and k give a comprehensive depiction of the BC mixing state," was imprecise. Our intended meaning was that " The characterization of the BC mixing state using the two indicators, χ and k, allows us to assess the steady-state phenomenon of the mixing state by examining both the aerosols composition and the distribution of the BC coating layer." We have added the related description in the revised manuscript (Page 5, Lines 143-145):

*"The characterization of the BC mixing state using the two metrics, χ and k, allows us to assess the steady-state phenomenon of the mixing state by examining both the aerosol composition and the distribution of the BC coating layer."*

Here, we offer a detailed explanation of the indicators χ and k. χ denotes the degree of mixing from external to internal between BC components and non-BC components. The parameter *k* characterizes the distribution of the coating layer thickness.

Consequently, if both $\chi$ and $k$ remain steady state, it can be inferred that the composition of BC and non-BC components, as well as the distribution of the BC coating layer, reach a steady state. This indicates that the mixing state is reaching a steady state.

**Specific comments:**

*1) Line 61 states that "key scientific questions remain such as determining the characteristic timescale to reach a steady state". The characteristic timescale has been examined here. Do the authors feel there are there remaining "key questions" to address that could be worth adding to this line?*

**Response:** Thank you for your reminder. In explanation to the "key questions," we have incorporated the following questions in Page 5, Lines 65-68:

*"(1) Do the properties of BC aerosols demonstrate a tendency towards a steady state, and how can the steady-state characteristic time be quantified? (2) Under the steady-state assumption, can the optical properties of BC be evaluated efficiently and accurately? (3) How to integrate the steady-state theory into models, and under what specific conditions is it applicable?"*

*2) Since Shannon entropy and particle diversity metrics are shown in Table 2, could how these enable a mixing state index please be briefly summarized in the text, which otherwise does not mention these?*

**Response:** Thank you for your correction. We have updated the expression in the text. [Page 6, Lines 157-170]

*"The entropy $H_i$ or diversity $D_i$ of an individual particle i quantifies the number of effective species within a particle. This metric spans from a minimum value, where $H_i = 0$ and $D_i = 1$, indicating a particle comprised solely of a single component (either BC or non-BC), to a maximum value, where $H_i = ln2$ and $D_i = 2$, signifying a particle composed of equal mass proportions of BC and non-BC components. Alpha diversity ($D_\alpha$) quantifies the average effective number of species per particle within a population, with values ranging from 1, indicating that each particle is composed of a single species (though not necessarily the same species across particles), to a maximum value of 2 when all particles exhibit identical mass fractions. Conversely, gamma diversity ($D_\gamma$) assesses the effective number of species within the entire population, with values spanning from 1, signifying a population consisting of a single species, to a maximum value when there are equal bulk mass fractions of all species present. The two population diversity indices, $D_\alpha$ (per-particle) and $D_\gamma$ (bulk) can be integrated to yield a single mixing state index $\chi$, which quantifies the homogeneity or heterogeneity of the population. This index spans from $\chi = 0$, indicating a fully externally mixed population where all particles are pure, to $\chi = 1$, signifying a fully internally mixed population where all particles exhibit identical mass fractions."*

*3) I find the Table 1 caption ("The related quantities calculated [...]") to be worded clunky. Perhaps this could more simply be described as "Metrics of particle mass" or otherwise rewritten?*

**Response:** Thank you for your suggestion. We agree that using "Metrics of particle mass" as the caption for Table 1 is more concise. [Page 5, Line 153]

*"Table 1. Metrics of species masses in particles adapted from Riemer and West, 2013"*

**4) There's an error in Table 2 where some of the instances of 'a' should instead be alphas (e.g. 'Ha', 'Da'), which gives an incorrect impression that these metrics are species-specific.**

**Response:** Thank you for your correction. We have updated the expression in Table 2. [Page 6, Line 171]

**"Table 2. The computation of diversity metrics, adapted from Riemer and West, 2013"**

| Quantity | Name | Meaning |
|---|---|---|
| $H_i = \sum\limits_{a=1}^{2} -p_i^a \cdot \ln p_i^a$ | Mixing entropy of particle $i$ | Shannon entropy of species distribution within particle $i$ |
| $H_\alpha = \sum\limits_{i=1}^{N} p_i \cdot H_i$ | Average particle mixing entropy | Shannon entropy of species distribution within particle $i$ |
| $H_\gamma = \sum\limits_{a=1}^{2} -p^a \cdot \ln p^a$ | Population bulk mixing entropy | Shannon entropy of species distribution within the population |
| $D_i = e^{H_i} = \prod\limits_{a=1}^{2} (p_i^a)^{-p_i^a}$ | Particle diversity of particle $i$ | effective number of species in particle $i$ |
| $D_\alpha = e^{H_\alpha} = \prod\limits_{i=1}^{N} (D_i)^{p_i}$ | Average particle (alpha) species $i$ diversity | average effective number of species in each particle |
| $D_\gamma = e^{H_\gamma} = \prod\limits_{a=1}^{2} (p^a)^{-p^a}$ | Bulk population (gamma) species diversity | effective number of species in the bulk |
| $\chi = \dfrac{D_\alpha - 1}{D_\gamma - 1}$ | Mixing state metric | degree to which population is externally mixed ($\chi = 0\%$) versus internally mixed ($\chi = 100\%$) |

**5) It seems a bit odd for there to be 2 tables full of aerosol parameters without including the 'k' that is used extensively in this study.**

**Response:** Thank you for your inquiry. Given the complexity involved in calculating the chi parameter, we have referenced prior studies by Fierce and Riemer, among others, and have synthesized the information into a tabular format for clarity and conciseness. Regarding the parameter $k$, which represents the slope of the exponential linear distribution, its determination is more straightforward. Consequently, we have included the detailed calculation in Section 2.1 of the Supplement. In the revised manuscript, we have rewritten the part on how to calculate the value of $k$ in Lines 173-187.

*"In this study, we also used the metric k adopted from our previous study (Wang et al., 2023) to quantify the CT distribution of BC aerosols during the simulation. Unlike metric χ, metric k*

*emphasizes the thickness of the BC coating layer and characterizes the mixing state of BC aerosols based on the CT distribution. The unified framework governing the mixing state of BC aerosols reveals a consistent size distribution, indicating that the natural logarithm of the number concentration (ln(n(CT))) and CT exhibit a linear relationship, regardless of the size of the BC core (Wang et al., 2023). Here, the variable $D_c$ denotes the diameter of the BC core and the variable $D_p$ denotes the diameter of the particle, CT signifies the coating thickness of the BC particle, which is defined as $D_p − D_c$. Furthermore, n(CT) represents the normalized number concentration of BC particles within the range of (CT − ΔCT/2, CT + ΔCT/2]. The normalization is relative to the total number concentration of BC particles, rendering n(CT) a dimensionless value. By performing linear regression, the relationship between the CT and the corresponding number concentration in logarithmic coordinates, ln(n(CT)), was established and the slope k was calculated by Eq. (1). The detailed methodology for data processing is provided in Sect. 2.1 in the Supplement.*

$$k = \frac{d\ ln(n(CT))}{d\ CT} \tag{1}$$

*Although the value of k can be calculated using Eq. (1), it is fundamentally determined by the deposition rate (Dep), or more generally, the removal rate, and the growth rate (GR), such that (k = $\frac{Dep}{GR}$) (Wang et al., 2023). The slope k was subsequently adopted as a characterization parameter to assess the BC mixing state, focusing on the particle size distribution."*

**6) The Results would be easier for the reader to follow if it were divided into a few sub-sections.**

**Response:** Thank you for your suggestion. We have now divided the results into the following four sub-sections.

*"3.1 The realism of the baseline case;3.2 The characteristic time of BC reaching a steady-state mixing state; 3.3 The exponential linear distribution of coating thickness and its reconciliation of non-uniform composition; 3.4 The application of the k-value method in BC absorption "*

**7) Lines 213-219 could be in the Methods, as they slightly distract from the flow of the results.**

**Response:** Thank you for your suggestion. We have put lines 213-219 into the Methods. Lines 217-224 in the revised manuscript.

*"**2.5 Determination of characteristic time***

*To quantify the rate at which the BC mixing state approaches a steady state, we employ the characteristic time τ, as defined by the following equation.*

$$metric(t) = metric\ (0) \cdot e^{-\frac{t}{\tau}} + metric(\infty) \cdot (1 - e^{-\frac{t}{\tau}}) \qquad (3)$$

*where the metric(0) denotes the mixing state at the initial state, while the metric($\infty$) denotes the steady state. In this study, we use metrix $\chi$ to characterize the mixing state. The steady-state characteristic time $\tau$ enables us to ascertain the timescale over which the mixing state reaches a steady state and facilitates a quantitative comparison of the differences across various cases. "*

**8)** *In Fig. 5 the label "Ideal uniform distribution of BC aerosols" seems off, since this is the coating thickness distribution, while BC is still lognormal if I'm not mistaken.*

**Response:** Thanks for the reminder. We've corrected the inaccuracies in the original Fig. 5 by adding color shading to represent the masses of the BC and non-BC components. [Page 14, Figure 4]

[Figure]

*"Figure 4. The reconciliation of the exponential linear CT distribution and the non-uniform composition of BC aerosols. (a) The exponential linear CT distribution of BC aerosols with a slope k of 0.016 nm⁻¹ under the steady state obtained from a PartMC-MOSAIC simulation. (b) The per-particle volume fraction of BC components and non-BC components with respect to per-particle BC mass. (c) The mass distribution of BC and non-BC components derived from the log-normal distribution of $D_c$ and the exponential linear distribution of CT (k = 0.016 nm⁻¹). (d) The mass distribution of aerosol components, normalized by dividing each component's mass by the total particle mass, with respect to the BC mass in each particle, was determined using the statistical method of Fierce et al. (2017)*

*based on the results obtained from the PartMC-MOSAIC simulation. The values on the y-axis in (c) and (d) have*

*been normalized* ($\frac{dM_i/dlog(m_{BC})}{\int_{m_{BC}=10^{-5}}^{m_{BC}=1} dM_i/dlog(m_{BC}) \cdot dlog(m_{BC})}$). *"*

**9)** *In the caption to Fig. 5, should "normalized by dividing each component's mass by the maximum particle mass" instead be "total particle mass" following the other normalization described below?*

**Response:** Thank you for your reminder. We have corrected the expression error here. [Page 15, Line 324]

*"(d) The mass distribution of aerosol components, normalized by dividing each component's mass by the total particle mass, with respect to the BC mass in each particle, was determined using the statistical method of Fierce et al. (2017)"*

**10)** *The Discussion material feels like it could be a Results subsection. It might make sense to instead use the Discussion for putting the findings into the broader context of other work and explain the potential utility for modeling efforts.*

**Response:** Thank you for your suggestion. We have put the discussion material into section 3.3 The exponential linear distribution of coating thickness and its reconciliation of uniform composition. We discussed how to put the findings into the broader context of other work and explain the potential utility for modeling efforts.

**References**

Bond, T. C., Doherty, S. J., Fahey, D. W., Forster, P. M., Berntsen, T., DeAngelo, B. J., Flanner, M. G., Ghan, S., Kärcher, B., Koch, D., Kinne, S., Kondo, Y., Quinn, P. K., Sarofim, M. C., Schultz, M. G., Schulz, M., Venkataraman, C., Zhang, H., Zhang, S., Bellouin, N., Guttikunda, S. K., Hopke, P. K., Jacobson, M. Z., Kaiser, J. W., Klimont, Z., Lohmann, U., Schwarz, J. P., Shindell, D., Storelvmo, T., Warren, S. G., and Zender, C. S.: Bounding the role of black carbon in the climate system: A scientific assessment, J. Geophys. Res. Atmospheres, 118, 5380–5552, https://doi.org/10.1002/jgrd.50171, 2013.

Chen, G., Wang, J., Wang, Y., Wang, J., Jin, Y., Cheng, Y., Yin, Y., Liao, H., Ding, A., Wang, S., Hao, J., and Liu, C.: An Aerosol Optical Module With Observation-Constrained Black Carbon Properties for Global Climate Models, J. Adv. Model. Earth Syst., 15, e2022MS003501, https://doi.org/10.1029/2022MS003501, 2023.

Chen, G., Liu, C., Wang, J., Yin, Y., and Wang, Y.: Accounting for Black Carbon Mixing State, Nonsphericity, and Heterogeneity Effects in Its Optical Property Parameterization in a Climate Model, J. Geophys. Res. Atmospheres, 129, e2024JD041135, https://doi.org/10.1029/2024JD041135, 2024.

Ding, A., Nie, W., Huang, X., Chi, X., Sun, J., Kerminen, V.-M., Xu, Z., Guo, W., Petäjä, T., Yang, X., Kulmala, M., and Fu, C.: Long-term observation of air pollution-weather/climate interactions at the SORPES station: a review and outlook, Front. Environ. Sci. Eng., 10, 1–15, https://doi.org/10.1007/s11783-016-0877-3, 2016.

Fierce, L., Bond, T. C., Bauer, S. E., Mena, F., and Riemer, N.: Black carbon absorption at the global scale is affected by particle-scale diversity in composition, Nat. Commun., 7, 12361, https://doi.org/10.1038/ncomms12361, 2016.

Liu, X., Ma, P.-L., Wang, H., Tilmes, S., Singh, B., Easter, R. C., Ghan, S. J., and Rasch, P. J.: Description and evaluation of a new four-mode version of the Modal Aerosol Module (MAM4) within version 5.3 of the Community Atmosphere Model, Geosci. Model Dev., 9, 505–522, https://doi.org/10.5194/gmd-9-505-2016, 2016.

Riemer, N. and West, M.: Quantifying aerosol mixing state with entropy and diversity measures, Atmospheric Chem. Phys., 13, 11423–11439, https://doi.org/10.5194/acp-13-11423-2013, 2013.

Riemer, N., West, M., Zaveri, R. A., and Easter, R. C.: Simulating the evolution of soot mixing state with a particle-resolved aerosol model, J. Geophys. Res., 114, D09202, https://doi.org/10.1029/2008JD011073, 2009.

Wang, J., Zhao, B., Wang, S., Yang, F., Xing, J., Morawska, L., Ding, A., Kulmala, M., Kerminen, V.-M., Kujansuu, J., Wang, Z., Ding, D., Zhang, X., Wang, H., Tian, M., Petäjä, T., Jiang, J., and Hao, J.: Particulate matter pollution over China and the effects

of control policies, Sci. Total Environ., 584–585, 426–447, https://doi.org/10.1016/j.scitotenv.2017.01.027, 2017.

Wang, J., Wang, J., Cai, R., Liu, C., Jiang, J., Nie, W., Wang, J., Moteki, N., Zaveri, R. A., Huang, X., Ma, N., Chen, G., Wang, Z., Jin, Y., Cai, J., Zhang, Y., Chi, X., Holanda, B. A., Xing, J., Liu, T., Qi, X., Wang, Q., Pöhlker, C., Su, H., Cheng, Y., Wang, S., Hao, J., Andreae, M. O., and Ding, A.: Unified theoretical framework for black carbon mixing state allows greater accuracy of climate effect estimation, Nat. Commun., 14, 2703, https://doi.org/10.1038/s41467-023-38330-x, 2023.

Zheng, Z., Curtis, J. H., Yao, Y., Gasparik, J. T., Anantharaj, V. G., Zhao, L., West, M., and Riemer, N.: Estimating Submicron Aerosol Mixing State at the Global Scale With Machine Learning and Earth System Modeling, Earth Space Sci., 8, https://doi.org/10.1029/2020EA001500, 2021.

---

## Author Comment (AC2)

**Response to the comments of Reviewer #2 (EGUSPHERE-2024-1924)**

*Reviewer #2: The manuscript "Steady-State Mixing State of Black Carbon Aerosols from a Particle-Resolved Model" by Zhang et al., investigates the mixing state of black carbon aerosols. Their results indicate, based on both a particle-resolved model and observations in Nanjing that the mixing state of BC aerosol reaches a steady state in few hours. The results of the paper are interesting and can be useful for improving the treatment of mixing of BC with scattering compounds in atmospheric models. However, the current study does not provide yet the means to apply this finding in models. The paper in within the scope of ACP, it presents novel ideas, reaches substantial conclusions. The paper is well written and I can recommend accepting it for publication after the following issues are addressed.*

**Response:** Thanks to reviewer #2 for the insightful suggestions and comments. In response, we have restructured the manuscript and have included detailed, point-by-point responses to all comments and questions raised. The reviewer's comments are presented in italics, followed by our responses and revisions highlighted in blue. In the revised manuscript, we have elaborated on the application methods and potential fields of application for the study's results in response to Main Comment 1. Additionally, we have clarified the description of the gas setup on Page 4, Lines 105-108, to provide a clearer introduction to the setup, addressing Main Comment 2. Furthermore, we have added legends explanations for curves and points in the caption of Figures S2 and S3, in response to Technical Comment 1. We contend that these modifications, prompted by your valuable feedback, have significantly enhanced the coherence and readability of the article. Here are our point-to-point responses.

***Main Comments:***

*1. My main comment is related to how to apply this knowledge in atmospheric modelling. I assume that since this holds only near emission sources, parameterizing the mixing state should be embedded in emission schemes, right? Away from the sources and higher up in the atmosphere where there are no emissions and surface removal, such steady state assumption may not hold. It would be good to add some discussion about this.*

**Response:** Thank you for your comment. We discuss the potential utility of steady-state theory for modeling efforts and its application scopes in the "4. Discussion" of the revised manuscript. To be brief, the characteristic timescale for BC approaching a steady-state mixing state is significantly shorter than its atmospheric lifetime, suggesting that the properties of BC under the steady state can effectively represent the averaged properties of BC across extensive spatial and temporal scales. The parameter $k$ denotes the distribution of coating thickness ($n(CT) = kN \cdot e^{-k \cdot CT}$) under steady state, obtained from the growth rate and removal rate, thus efficiently helping calculate the light absorption capacity of BC aerosols. Regarding the application of the slope $k$ in models, we propose utilizing existing state values in models, such as the total mass concentration of each component combined with the distribution of coating thickness and diameter of BC-core, to build a parameterized framework of the $k$ value based on the studies by Chen et al. (2023, 2024), which is an avenue for future research. Furthermore, we plan to employ machine learning techniques to develop an emulator for the $k$ value based on training data obtained from PartMC-MOSAIC. In this approach, the $k$ value will serve as the "label", while emissions, initial conditions, and meteorological conditions will be treated as "features". These tasks are avenues for our future work.

We appreciate your interpretation of the applicability of the steady-state assumption, particularly the point that it 'holds only near emission sources, and parameterizing the mixing state should be embedded in emission schemes.' In the revised manuscript, we have provided a detailed interpretation to fully capture the nuances of the assumption's applicability in various conditions. To be more precise, the steady-state assumption can be applied to the analysis of the average properties of BC across extensive spatial and temporal scales. According to the region away from the sources and higher up in the atmosphere where there are no emissions and surface removal, t the BC mixing state is considered to have reached a steady state. In determining the $k$ value, the emissions and removal of BC aerosols can be represented by their transport dynamics, including both inflow and outflow, within the region.

We have added the description of the application of our findings in models in the revised manuscript (Lines 348-370 ):

*"We discuss the potential utility of steady-state theory for modeling efforts and its application scopes. The slope parameter k denotes the CT distribution of BC aerosols n(CT) = kN · e^{-k · CT} under steady state. According to the derivation presented by Wang et al. (2023), the value of 1/k was*

*determined to be equivalent to the average CT of BC aerosols (Table S3). The relationship between $E_{abs}$ and CT is approximately linear when CT is less than 200 nm (k > 0.005 nm$^{-1}$) (demonstrated in the Fig. S3 of Wang et al., 2023). Therefore, a monodisperse CT, 1/k, can replace the BC coating thickness distribution when calculating the BC absorption. Fig. 5 in this paper shows that the $E_{abs}$ based on the steady-state theory can serve as a characterization of the light absorption enhancement of BC aerosols under the steady-state condition. The k-value can be obtained from the growth rate and removal rate (k = Dep/GR), thus efficiently helping evaluate the BC absorption. Regarding the application of the slope k in models, we propose utilizing existing state values in models, such as the total mass concentration of each component combined with the distribution of coating thickness and diameter of BC-core, to build a parameterized framework of the k value based on the studies by Chen et al. (2023, 2024), which is an avenue for future research. Furthermore, we plan to employ machine learning techniques to develop an emulator for the k value based on training data obtained from PartMC-MOSAIC. In this approach, the k value will serve as the 'label', while emissions, initial conditions, and meteorological conditions will be treated as 'features'. This method is inspired by the study by Zheng et al. (2021), which used the matrix χ as the 'label' for their emulator. Further, the characteristic timescale for BC reaching a steady-state mixing state ranges from 1.9 to 9.7 hours, which is considerably shorter than their atmospheric lifetime, typically around 7 days. Thus, the steady-state assumption may effectively be applied to climate models, which are generally concerned with the characteristics of BC aerosols across extensive spatial and temporal scales. While it may function less well for high-resolution modeling studies of urban areas wherein a larger percent of BC emissions are fresh, in which the steady state may be influenced. In these conditions, we propose categorizing BC-containing particles into two distinct types: fresh BC, which lacks a coating layer, and aged BC, which possesses a coating layer. The slope parameter k is capable of characterizing the distribution of the latter. Hence, the k-value method can be applied in models for BC absorption under the steady-state condition."*

*2. Page 3, Line 97: "In this study, the initial gas concentration and emission rate were slightly adjusted based on Riemer et al. (2009) according to Wang et al. (2017)." The meaning of the sentence is unclear. What was the initial assumption for gas concentrations and emissions and how were they adjusted? Were these assumption tuned to match the model with observations?*

**Response:** Thank you for your reminder. The previous sentence may be deemed somewhat misleading. We employed the gas setup of Riemer et al. (2009) within PartMC, as Riemer et al., being the developers and experienced users of PartMC, have established a gas setup that is both authoritative and widely adopted in subsequent. To better reflect the aerosol composition characteristic of Chinese conditions, we subsequently adjusted the gas setup based on the field observations in China by Wang et al. (2017). To more precisely communicate our intended message, we have revised the original statement as follows: [Page 4, Lines 105-108]

*"In this study, the initial gas concentration and emission rate were established in accordance with the parameters set by Riemer et al. (2009). To reflect the typical composition of aerosols in China, the setup for the baseline case was subsequently adjusted based on the observations reported by Wang et al. (2017), as detailed in Table S1 of the Supplement."*

*"**Table S1.** Input variables assigned in the baseline case"*

| Environmental variable | Value |
|---|---|
| Temperature [K] | 289 |
| Relative humidity | 0 |
| Boundary layer height [m] | 293.14 |
| Mass loss (deposition) | constant |
| Latitude | 0 °N |
| Day of year | July 19 |
| **Aerosol characteristic** | **Value** |
| Emission rate [$m^{-2}\,s^{-1}$] | $1.8 \times 10^7$ |
| Fraction Bare-BC emissions | 24.6% |
| Fraction Mix-BC emissions | 2.4% |
| Fraction BC-free emissions | 72.7% |
| **Aerosol type** | **Geo. mean dia. [nm]** |
| Bare-BC | 89 |
| Mix-BC | 109 |
| BC-free | 110 |
| **Aerosol type** | **Geo. standard dev.** |
| Bare-BC | 1.60 |
| Mix-BC | 1.60 |
| BC-free | 1.70 |
| **Aerosol type** | **Mass composition** |
| Bare-BC | 100% BC |
| Mix-BC | 68.3% BC, 31.7% OC |
| BC-free | 100% OC |
| **Emitted gas species** | **Rate [$mol \cdot m^{-2} \cdot s^{-1}$]** with a total multiplication factor of 25 % |
| Sulfur dioxide | $2.51 \times 10^{-8}$ |
| Nitrogen dioxide | $1.20 \times 10^{-9}$ |
| Nitric oxide | $2.50 \times 10^{-8}$ |
| Ammonia | $6.11 \times 10^{-9}$ |
| Carbon monoxide | $2.91 \times 10^{-7}$ |
| Acetaldehyde | $6.80 \times 10^{-10}$ |
| Formaldehyde | $1.68 \times 10^{-9}$ |
| Ethene | $7.20 \times 10^{-9}$ |
| Internal olefin carbons | $2.42 \times 10^{-9}$ |
| Terminal olefin carbons | $2.42 \times 10^{-9}$ |
| Toluene | $4.04 \times 10^{-9}$ |
| Xylene | $2.41 \times 10^{-9}$ |

| | |
|---|---|
| Acetone | $1.23 \times 10^{-9}$ |
| Paraffin carbon | $9.60 \times 10^{-8}$ |
| Isoprene | $2.30 \times 10^{-10}$ |
| Methanol | $2.80 \times 10^{-10}$ |
| Alcohols | $3.45 \times 10^{-9}$ |

*3. Was the motivation for the additional cases to show that for all conditions, the exponential linear distribution occurs?*

**Response:** Thank you for your question. Yes, we aim to convey that the exponential linear distribution universally applies (occurs in all conditions), although the additional cases may not encompass all conditions. To improve the generalizability of our findings, we varied the temperature, the size distribution of freshly emitted BC, and the particle-gas emission ratio, constructing additional cases. We can simplify the calculation of BC absorption by applying the exponential linear distribution. The exponential linear distribution of CT can be represented by a single slope parameter $k$ ($n(\text{CT}) = kN \cdot e^{-k \cdot \text{CT}}$). According to the derivation presented by Wang et al. (2023), the value of $1/k$ was determined to be equivalent to the average CT of BC aerosols (Table S3). The relationship between $E_{abs}$ and CT is approximately linear when CT is less than 200 nm ($k > 0.005$ nm$^{-1}$). This linear behavior is clearly illustrated in Fig. 1, which corresponds to Figure S3 in Wang et al. (2023). Therefore, a monodisperse CT, $1/k$, can replace the BC coating thickness distribution when calculating the BC absorption.

[Figure]

Figure 1. Change of mass absorption cross-section (MAC) of black carbon (BC) with coating thickness (ΔDp). Blue dots represent the calculated MAC based on core-shell Mie theory with 5 the linear fit shown as the red line.

Besides, we have incorporated three additional cases into the revised manuscript to address the previously insufficient representation of natural biomass burning black carbon, as pointed out by Reviewer #1.

The geometric mean diameter of freshly emitted BC was set as 150 nm based on Fig. 2, which corresponds to Figure 4 of Bond et al.(2013), and varying particle-gas emission ratios. In these three new scenarios, the CT distribution follows an exponential linear distribution, with a steady-state timescale ranging from 2.1 to 2.8 hours.

[Figure]

Figure 2. Mass and number size distributions of BC particles observed in three fresh urban (red) and two fresh biomass burning (black) plumes as identified in the legend. The measurements are made on board an aircraft using an in situ, single-particle detection instrument (SP2). The variable coatings on the BC particles are not shown. The observed (a) mass and (b) number amounts are plotted as symbols versus volume equivalent diameter based on assuming a spherical particle shape. The mass distributions are normalized to the same peak value. The observations are fit by a lognormal function between 90 and 600 nm (solid lines). The number distribution fits are those consistent with the fit to the respective mass distribution and are scaled to represent the same BC mass. From Schwarz et al. [2008b].

The three new cases are detailed in Sect. 1.2 of the Supplement, with their CT distributions shown in Figures S3 (j), (k), and (l), and steady-state characteristic timescales listed in Table S3 of Supplement.

[Figure]

*"Part of Fig S3. Results of CT distribution of BC aerosols in 12 simulation cases (excluding the Baseline case)."*

*"Part of Table S3. The characteristic time τ, equivalent CT (1/k), coefficient of determination of linear regression, and the average true CT value for ten different cases."*

| Scenario | Characteristic time (hour) | Equivalent CT (nm) | Coefficient of determination | Average true CT (nm) |
|---|---|---|---|---|
| Baseline case | 3.2 | 62 | 0.999 | 63 |
| Case 10 | 2.8 | 29 | 0.995 | 29 |
| Case 11 | 2.1 | 71 | 0.999 | 76 |
| Case 12 | 2.2 | 157 | 0.998 | 162 |

**Technical comment:**

*1. Please add legends or explanations for curves and points in Figures S2 and S3*

**Response:** Thank you for your reminder. We have added legends explanations for curves and points in the caption of Figures S2 and S3. [Page 5 and 7]

[revised manuscript text omitted]